# Plantainoside D Reduces Depolarization-Evoked Glutamate Release from Rat Cerebral Cortical Synaptosomes

**DOI:** 10.3390/molecules28031313

**Published:** 2023-01-30

**Authors:** Kuan-Ming Chiu, Ming-Yi Lee, Cheng-Wei Lu, Tzu-Yu Lin, Su-Jane Wang

**Affiliations:** 1Division of Cardiovascular Surgery, Cardiovascular Center, Far-Eastern Memorial Hospital, New Taipei City 22060, Taiwan; 2Department of Electrical Engineering, Yuan Ze University, Taoyuan 32003, Taiwan; 3Department of Medical Research, Far-Eastern Memorial Hospital, New Taipei City 22060, Taiwan; 4Department of Anesthesiology, Far-Eastern Memorial Hospital, New Taipei City 22060, Taiwan; 5Department of Mechanical Engineering, Yuan Ze University, Taoyuan 32003, Taiwan; 6School of Medicine, Fu Jen Catholic University, New Taipei City 24205, Taiwan; 7Research Center for Chinese Herbal Medicine, College of Human Ecology, Chang Gung University of Science and Technology, Taoyuan 33303, Taiwan

**Keywords:** plantainoside D, glutamate release, voltage-dependent Ca^2+^ channel, PKC, cerebral cortex, synaptosomes

## Abstract

Inhibiting the excessive release of glutamate in the brain is emerging as a promising therapeutic option and is efficient for treating neurodegenerative disorders. The aim of this study is to investigate the effect and mechanism of plantainoside D (PD), a phenylenthanoid glycoside isolated from *Plantago asiatica* L., on glutamate release in rat cerebral cortical nerve terminals (synaptosomes). We observed that PD inhibited the potassium channel blocker 4-aminopyridine (4-AP)-evoked release of glutamate and elevated concentration of cytosolic Ca^2+^. Using bafilomycin A1 to block glutamate uptake into synaptic vesicles and EDTA to chelate extracellular Ca^2+^, the inhibitory effect of PD on 4-AP-evoked glutamate release was prevented. In contrast, the action of PD on the 4-AP-evoked release of glutamate in the presence of dl-TBOA, a potent nontransportable inhibitor of glutamate transporters, was unaffected. PD does not alter the 4-AP-mediated depolarization of the synaptosomal membrane potential, suggesting that the inhibitory effect of PD on glutamate release is associated with voltage-dependent Ca^2+^ channels (VDCCs) but not the modulation of plasma membrane potential. Pretreatment with the Ca^2+^ channel blocker (N-type) ω-conotoxin GVIA abolished the inhibitory effect of PD on the evoked glutamate release, as did pretreatment with the protein kinase C inhibitor GF109203x. However, the PD-mediated inhibition of glutamate release was eliminated by applying the mitochondrial Na^+^/Ca^2+^ exchanger inhibitor CGP37157 or dantrolene, which inhibits Ca^2+^ release through ryanodine receptor channels. These data suggest that PD mediates the inhibition of evoked glutamate release from synaptosomes primarily by reducing the influx of Ca^2+^ through N-type Ca^2+^ channels, subsequently reducing the protein kinase C cascade.

## 1. Introduction

The secretion and reception of neurotransmitters is a principal mechanism by which neurons communicate with other neurons or target cells. Glutamate, which activates postsynaptic cells, is the primary excitatory neurotransmitter of the vertebrate central nervous system. As the most abundant free amino acid, glutamate is implicated in many brain functions, such as normal neural transmission, plasticity, learning and memory [1,2,3]. When excess glutamate is released into the extracellular space, the excessive activation of glutamate receptors occurs as well as the uncontrolled continuous depolarization of neurons, which is a process called excitotoxicity. Excessive glutamate allows high levels of calcium ions (Ca^2+^) to enter the cell and disrupt calcium homeostasis. This process causes many deleterious consequences, including oxidative stress, mitochondrial dysfunction, and the release of lysosomal enzymes, ultimately leading to the deterioration of neuronal function and loss of neurons [4,5]. Neuronal death caused by excitotoxicity is known to be associated with numerous degenerative nerve disorders, including Parkinson’s disease, Huntington’s disease and Alzheimer’s disease [6]. Thus, decreasing the amount of glutamate released from nerve terminals is a prospective neuroprotective approach for neurological conditions in which pathologies are associated with excitotoxicity [7].

Plantainoside D (PD) is an active component of the phenylenthanoid glycoside class widely distributed in a variety of natural plants, such as *Plantago asiatica* L., *Digitalis purpurea*, *Picrorhiza Scrophulariiflora*, *Chirita longgangensis var. hongyao* and *Lagotis integra* W. W. Smith [8,9,10,11,12]. Previous studies have shown that PD exhibits a series of impressive pharmacological effects, such as inhibitory activity against angiotensin-converting enzyme, an anti-renal fibrosis effect, cardioprotection, anti-inflammatory activity, and antioxidative activity [13,14,15,16,17]. Compounds with anti-inflammatory and antioxidative properties might confer neuroprotective effects, considering that oxidative stress and neuroinflammation are essential elements of neuronal death in various neurodegenerative disorders [18,19]. PD has been indicated to exhibit strong antioxidative activity and scavenging effects on superoxide anion radicals. PD exerts antiapoptotic action by suppressing the generation of reactive oxygen species (ROS) and the activation of nuclear factor-κB kinase, which is a central mediator of the proinflammatory signaling pathway [14]. Moreover, PD shows inhibitory activity against PKC-α [20,21], which plays a major role in regulating terminal depolarization and transmitter glutamate release. These results imply that PD may be a potential neuroprotective agent against neurotoxicity induced by the excessive release of glutamate.

Isolated nerve terminals (synaptosomes) are a widely used system to study neural transmission and synaptic dysfunction in vitro, and with this system, pharmacological and biochemical analyses of presynaptic effects are performed without interference from postsynaptic neurons. Therefore, this study uses synaptosomes to assess the effect of PD on glutamate release as well as on the molecular mechanism underlying PD action. Moreover, the membrane potential in synaptosomes, cytosolic free calcium concentration ([Ca^2+^]c), the downstream modulation of voltage-dependent Ca^2+^ channels (VDCCs) and the phosphorylation of protein kinase are monitored to assess the efficacy of PD on depolarization-evoked glutamate release.

## 2. Results

### 2.1. Inhibitory Effects of Plantainoside D on 4-AP-Induced Glutamate Release from Rat Cortical Synaptosomes

In this study, the ability of PD to influence the evoked release of glutamate from synaptosomal fractions of rat cortical synaptosomes was assessed. The extracellular level of glutamate was monitored online continuously using an enzyme-linked fluorometric assay [22]. The potassium channel blocker 4-aminopyridine (4-AP) was used in this study, as it has been reported that 4-AP can activate voltage-dependent Ca^2+^ channels (VDCCs) and stimulate the release of vesicular glutamate from nerve endings. The experimental design and treatment schedule are shown in Figure 1. In the presence of CaCl_2_, depolarization with 1.2 mM 4-AP induced glutamate release to 7.8 ± 0.1 nmol mg of protein^−1^ 5 min^−1^ (Figure 2A). However, adding 30 µM PD prior to the 4-AP addition did not affect the release of basal glutamate but substantially reduced the evoked glutamate release to 4.2 ± 0.6 nmol mg of protein^−1^ 5 min^−1^ (*n* = 5; *p <* 0.0001). This dose–response curve was obtained with the preincubation of synaptosomes for 10 min using different concentrations of PD (Figure 2B). The inhibitory effects of PD on 4-AP-induced glutamate release are evident down to 5 μM, while maximum inhibition occurs at 50 μM. The half-maximal inhibitory concentration (IC 50) value for PD was 32 μM.

### 2.2. PD Inhibition of 4-AP-Induced Glutamate Release Is Mediated by an Exocytotic Mechanism

Two mechanisms have been suggested to modulate the 4-AP-induced release of glutamate from synaptosomes: Ca^2+^-dependent and Ca^2+^-independent release. In the Ca^2+^-dependent exocytosis pathway, glutamate is released by the fusion of synaptic vesicles with the plasma membrane. Another relevant mechanism is the reversal efflux of glutamate by the glutamate transporter, which is Ca^2+^-independent [23,24]. As shown in Figure 3A, experiments in an extracellular Ca^2+^-free solution containing 300 μM EGTA [25] demonstrated that 1.2 mM 4-AP stimulated 1.4 ± 0.2 nmol mg of protein^−1^ 5 min^−1^ glutamate. However, this Ca^2+^-independent release was not affected by PD at 30 μM (1.5 ± 0.3 nmol mg of protein^−1^ 5 min^−1^; *p* = 0.9). Moreover, bafilomycin A1, a pharmacological tool that prevents the uptake and storage of glutamate in synaptic vesicles [26], or dl-threo-beta-benzyl-oxyaspartate (dl-TBOA), a useful tool for investigating the physiological roles of transporters, was used to study the effect of PD on glutamate release (F (1.004, 35.15) = 33.64, *p <* 0.0001, Figure 3B). At a concentration of 0.1 μM, the specific vacuolar ATPase inhibitor bafilomycin A1 reduced 4-AP-induced glutamate release to 1.6 ± 0.2 nmol mg of protein^−1^ 5 min^−1^. Compared with the bafilomycin A1 treatment alone, the application of 30 μM PD did not effectively change the glutamate release induced by 1.2 mM 4-AP (1.2 ± 0.1 nmol mg of protein^−1^ 5 min^−1^, *p* = 0.1, Figure 3B). Conversely, the nonselective glutamate transporter inhibitor dl-TBOA (10 μM) caused a significant increase in 4-AP-induced glutamate release (13.7 ± 0.5 nmol mg of protein^−1^ 5 min^−1^, *p <* 0.0001). However, our data show that PD still effectively decreased 4-AP-induced glutamate release in the presence of dl-TBOA. Together, these data indicate that PD regulates glutamate release via Ca^2+^-dependent synaptic vesicular exocytosis rather than Ca^2+^-independent cytosolic glutamate efflux mediated by transporters.

### 2.3. Effect of PD on Cytosolic [Ca^2+^] and Synaptosomal Membrane Potential

To explore the underlying mechanisms responsible for the PD-mediated regulation of glutamate release, cytosolic Ca^2+^ levels were monitored using the fluorescent dye Fura-2 (5 μM). In rat cortical synaptosomes, we observed an average increase of 255.2 ± 14.0 nM cytosolic [Ca^2+^] by adding 1.2 mM 4-AP (from 209.9 ± 14.0 nM to a plateau of 465.1 ± 22.9 nM). Under basal conditions and in the presence of 1 mM extracellular Ca^2+^, PD did not significantly alter their resting Ca^2+^ levels (194.1 ± 10.3 nM). Moreover, PD reduced 4-AP-induced cytosolic [Ca^2+^] to 114.6 ± 9.0 nM (from 194.0 ± 10.3 nM to a plateau of 308.6 ± 8.0 nM; *p <* 0.0001; Figure 4A). In addition, the effect of PD on the synaptosomal plasma membrane potential was measured using 5 μM DiSC3(5), a cationic membrane-permeable fluorescent dye that is potential-sensitive. With the addition of 1.2 mM 4-AP, an increase in DiSC3(5) fluorescence from 1.3 ± 1.5 fluorescence units mg of protein^−1^ 5 min^−1^ to 22.3 ± 1.0 fluorescence units mg of protein^−1^ 5 min^−1^ was observed. Pretreatment with PD (30 μM) for 10 min caused no effect on the resting membrane potential or the 4-AP-induced increase in DiSC3(5) fluorescence (22.3 ± 1.0 fluorescence units mg of protein^−1^ 5 min^−1^; *p* = 0.7; Figure 4B). Furthermore, the effect of DHC on glutamate release under basal or depolarized (15 mM KCl) conditions was evaluated. The application of PD (30 μM) significantly decreased KCl (15 mM)-induced glutamate release compared with that of the control group (*p* < 0.0001, Figure 5); this process is Ca^2+^-channel-dependent and Na^+^-channel-independent [27]. These data provide supportive evidence that the inhibitory effect of PD on glutamate release is associated with VDCCs but not the modulation of plasma membrane potential.

### 2.4. Effects of Ca^2+^-Channel Antagonists on PD-Mediated Inhibition of 4-AP-Induced Glutamate Release

To better understand the mechanism underlying the inhibitory effect of PD on 4-AP-induced glutamate release and increase in Ca^2+^ influx, Ca^2+^-channel antagonists were used in this study. Previous studies have suggested that glutamate release is mainly dependent on Ca^2+^ influx through N- and P/Q-type VDCCs [28,29]. In rat cortical synaptosomes, control glutamate release evoked by 4-AP was 7.4 ± 0.3 nmol mg of protein^−1^ 5 min^−1^. ω-conotoxin GVIA (500 nM) and ω-agatoxin IVA (100 nM), which selectively blocked N- and P/Q-type VDCCs, respectively, reduced 4-AP-induced glutamate release to 4.1 ± 0.5 and 3.4 ± 0.7 nmol mg of protein^−1^ 5 min^−1^, respectively. The effects of PD and ω-conotoxin GVIA or ω-agatoxin IVA combinations on 4-AP-evoked glutamate release are presented in Figure 6A (F (1.008, 35.27) = 32.89, *p <* 0.0001). Together with ω-agatoxin IVA, PD still exerted an inhibitory effect on 4-AP-induced glutamate release (1.7 ± 0.4 nmol mg of protein^−1^ 5 min^−1^, *p <* 0.0001). However, the inhibitory action was abolished by the combined application of PD and ω-conotoxin GVIA (3.8 ± 0.5 nmol mg of protein^−1^ 5 min^−1^; *p* = 0.1).

Furthermore, Ca^2+^ released from intracellular stores, such as mitochondria and the endoplasmic reticulum, is also involved in glutamate release. Dantrolene and CGP37157, which prevent ER and mitochondrial calcium efflux, were used to investigate the involvement of intracellular calcium in the inhibitory effect of PD on glutamate release. In the control group, treatment with 1.2 mM 4-AP resulted in a vast increase in glutamate release to 7.1 ± 0.2 nmol mg of protein^−1^ 5 min^−1^. As shown in Figure 6B (F (1.010, 35.36) = 33.16, *p <* 0.0001), 10 μM dantrolene and 10 μM CGP37157 reduced 4-AP-induced glutamate release to 4.6 ± 0.4 and 4.4 ± 0.5 nmol mg of protein^−1^ 5 min^−1^, respectively. However, with the application of PD, the inhibitory effect on glutamate release was retained in the presence of dantrolene (*p <* 0.0001) or CGP37157 (*p <* 0.0001).

### 2.5. The Protein Kinase C (PKC) Pathway Is Responsible for the PD-Mediated Inhibition of Glutamate Release

Protein phosphorylation is an important cellular regulatory mechanism for the function of many components involved in glutamate release. We next investigated which protein kinase might account for the reduction in glutamate release mediated by PD. As seen in Figure 7, the protein kinase A (PKA) inhibitor *N*-[2-(pbromocinnamylamino)- ethyl]-5-isoquinolinesulfonamide dihydrochloride (H89), MAPK/ERK inhibitor 2-(2-amino-3-methoxyphenyl)-4*H*-1-benzopyran-4-one (PD98059) or protein kinase C (PKC) inhibitor bisindolylmaleimide I (GF109203X) reduced 1.2 mM 4-AP-induced glutamate release from 7.3 ± 0.4 nmol mg of protein^−1^ 5 min^−1^ to 3.1 ± 0.7 nmol mg of protein^−1^ 5 min^−1^ (*p <* 0.0001), 2.9 ± 0.3 nmol mg of protein^−1^ 5 min^−1^ (*p <* 0.0001) or 2.2 ± 0.1 nmol mg of protein^−1^ 5 min^−1^ (*p <* 0.0001). The application of PD still produced an inhibitory effect on 4-AP-induced glutamate release in the presence of the PKA inhibitor H89 (1.7 ± 0.3 nmol mg of protein^−1^ 5 min^−1^; *p <* 0.005) or the MAPK/ERK inhibitor PD98059 (1.6 ± 0.3 nmol mg of protein^−1^ 5 min^−1^; *p <* 0.01), whereas the inhibition was abolished by the protein PKC inhibitor GF109203X (2.3 ± 0.2 nmol mg of protein^−1^ 5 min^−1^; *p* = 0.9). These results suggest that the PKC pathway is responsible for the PD-mediated inhibition of glutamate release.

### 2.6. Effect of PD on Synaptosomal Protein Phosphorylation

To confirm that the action of PD on 4-AP-induced glutamate release is associated with the PKC pathway, we analyzed the phosphorylation of PKC-α in rat cortical synaptosomes. As shown in Figure 8, the relative levels of PKC-α phosphorylation in cortical synaptosomes stimulated with 4-AP were significantly higher than those in the control group (239.2 ± 53.7%; *p <* 0.05). The increase in PKC-α phosphorylation observed in the 4-AP alone group was significantly reduced by the PD treatment (54.1 ± 9.7%; *p <* 0.01). To confirm that the PKC pathway is involved in the action of PD, we performed experiments to test the effect of PD on the phosphorylation of synaptosomal-associated protein of 25 kDa (SNAP-25), a component of the SNARE complex and target for PKC [30,31]. Similar results were observed for SNAP-25, and the increased phosphorylation of SNAP-25 at Ser^187^ induced by 1.2 mM 4-AP treatment (228.9 ± 45.9%; *p <* 0.05) was also reduced by the application of PD (89.7 ± 27.7%; *p <* 0.05).

## 3. Discussion

Glutamate is a physiologically important excitatory neurotransmitter in the nervous system. Several studies have shown that the excessive release of glutamate in the extracellular space overexcited the receiving nerve cells, leading to the dysregulation of Ca^2+^ homeostasis, oxidative stress, cell damage and ultimately neuronal death [32,33]. This process is called excitotoxicity, which is the main cause of the pathogenesis of both acute and chronic neurologic diseases [4,34]. Thus, reducing excess levels of glutamate in the synaptic cleft is a potential strategy for preventing and treating neurologic diseases [35,36]. Since oxidative stress and neuroinflammation are common pathologic signatures of neurodegeneration, PD might possess neuroprotective effects, as PD exhibits a series of promising therapeutic potential, such as anti-inflammatory and antioxidative activity [13,14,15,16,17]. In this work, we observed that PD inhibited the 4-AP-evoked release of glutamate in association with Ca^2+^-dependent synaptic vesicular exocytosis and VDCCs rather than by modulating the plasma membrane potential. Our study revealed for the first time that PD exhibits an inhibitory effect on glutamate release from cortical synaptosomes.

In the present study, we examined the effect of PD on synaptosome preparation, a model that has been utilized since the 1960s and is excellent for studying synaptic physiology. Using the potassium channel blocker 4-AP as stimulator, an apparent inhibitory effect of PD on glutamate release in synaptosomes was observed. Our data show that PD inhibited glutamate release in a concentration-dependent manner, with an IC50 value of 32 μM after 10 min of pretreatment. The effects observed during 4AP-evoked glutamate release result from the following different mechanisms: (i) glutamate released via Ca^2+^-dependent vesicular exocytosis, in which both external Ca^2+^ influx and Ca^2+^ release from internal stores are involved. (ii) Ca^2+^-independent glutamate efflux via the reversal of a family of glutamate transporter proteins located on the plasma membrane. In a Ca^2+^-free medium, EGTA was added to each preparation together with PD, followed after 10 min by the addition of 4-AP. PD did not result in a significant decrease in Ca^2+^-independent glutamate release, which depends solely on the membrane potential [37]. Our results show that extracellular Ca^2+^ is necessary for the inhibitory effect of 4-AP-evoked glutamate release mediated by PD and that this effect is abolished when extracellular Ca^2+^ is removed. Using bafilomycin A1 to block glutamate uptake into synaptic vesicles, the inhibitory effect of PD on 4-AP-evoked glutamate release was prevented. In contrast, the action of PD on the 4-AP-evoked release of glutamate was unaffected in the presence of dl-TBOA, a potent nontransportable inhibitor of glutamate transporters. Together, PD inhibited the release of glutamate evoked by 4-AP from synaptosomes by decreasing the Ca^2+^-dependent exocytotic release of glutamate but not the reverse operation of glutamate transporters.

For exocytotic glutamate release, the predominant sources of cytoplasmic Ca^2+^ are intracellular Ca^2+^ released from internal stores and Ca^2+^ influx from the extracellular space mediated by VDCCs in the plasma membrane [38,39]. Dantrolene and CGP37157, which inhibit intracellular calcium flux across the sarcoplasmic reticulum and mitochondria, show no effect on PD-mediated inhibition of glutamate release. This implied that extracellular Ca^2+^ influx was involved in the action of PD rather than intracellular Ca^2+^ release from internal stores. Previously, it was reported that 4-AP-evoked glutamate release is associated with Na^+^ and Ca^2+^ channel activation, while KCl-stimulated glutamate release depends solely on Ca^2+^ channels [40,41]. Both 4-AP- and KCl-evoked glutamate release were significantly inhibited by PD, suggesting that the inhibitory effect of PD on glutamate release involved Ca^2+^ channels but not Na^+^ channels. To determine the involvement of synaptosomal plasma membrane potential in the PD-mediated inhibition of glutamate release, a membrane potential-sensitive dye, DiSC3(5), was used [42]. The results reveal that PD showed no effect on the resting or 4-AP depolarized membrane potential, suggesting that the inhibitory effect of PD on glutamate release is not correlated with increased K^+^ conductance. Additionally, PD significantly decreased the 4-AP-induced increase in [Ca^2+^]c, indicating that the action of PD is mediated by reducing extracellular influx through VDCCs. While the P/Q-type Ca^2+^ channel blocker ω-agatoxin IVA does not change the action of PD, the pretreatment with the N-type Ca^2+^ channel antagonist ω-conotoxin GVIA completely abolishes the PD-mediated inhibition of glutamate release. Taken together, these data suggest that the action of PD is mediated by Ca^2+^ influx through N-type Ca^2+^ channels, which regulates exocytosis and synaptic transmission in presynaptic nerve terminals.

The following possible mechanisms can explain the effect of PD on N-type Ca^2+^ channels: direct inhibition and indirect action through interactions with proteins located in the presynaptic terminal [43]. A number of protein kinases, including PKA, PKC and MAPK/ERK, likely influence the activity of presynaptic VDCCs, which subsequently alters neurotransmitter release [38,44]. To determine the protein kinase involved in the action of PD, the MAPK/ERK inhibitor PD98059, PKA inhibitor H89 and PKC inhibitor GF109203X were used in this study. Our results show that the PKC inhibitor prevented the inhibitory effect of PD on glutamate release, while the inhibitors of MAPK and PKA were ineffective. These findings collectively support that the PKC pathway plays a pivotal role in the PD-induced inhibitory effect on the release of glutamate evoked by 4-AP. Furthermore, we also demonstrated that PD significantly decreased the 4-AP-induced phosphorylation of PKC-α, a major classical Ca^2+^-dependent PKC isoform highly expressed in cerebral cortical synaptosomes that contributes to the regulation of terminal depolarization and the release of glutamate [37,45]. Previous studies have shown that PD exhibits inhibitory activity against PKC-α, which supports our findings [20,21]. However, in addition to PKC-α, the possible involvement of other PKC isoforms could not be ruled out. For instance, PKC-β (β1 and β2) has been reported to be involved in many different cellular functions in a Ca^2+^-dependent manner [46]. The synaptic protein SNAP-25 is a target of PKC and a member of the SNARE family of vesicular fusion proteins. PKC phosphorylation of residue serine 187 (Ser 187) [31,47] in SNAP-25 is important for assembly of the SNARE complex and Ca2+-triggered synaptic vesicle exocytosis [30,48]. 4-AP-induced SNAP-25 phosphorylation at Ser 187 was abolished by PD, suggesting the involvement of the synaptic protein SNAP-25 in the inhibitory effect of PD on glutamate release. In our previous study, no significant difference was observed in total proteins of PKC-α and SNAP25 between the groups [22].

In this study, the ability of PD to decrease glutamate release is dose-dependent with an IC50 value of 32 μM. Consistent with our study, 20 μg/mL PD inhibits cardiac muscle cell apoptosis [14]. In addition, in an in vivo animal study, PD was administered intravenously (2.0 mg/kg) and orally (10 mg/kg) to rats for pharmacokinetic analysis [11]. They found that the single-dose oral administration of PD results in a low bioavailability in vivo. Since there are very few in vivo studies of PD in literature, further studies are needed to understand the dosage for the treatment of neurodegenerative diseases.

In conclusion, the inhibitory effect of PD on 4-AP-evoked glutamate release can be useful for the justification of this compound in the treatment of neurological disorders linked to excitotoxicity. Suppressing Ca^2+^ influx from the extracellular space into the presynaptic terminals through N-type Ca^2+^ channels contributed to the inhibitory effect of PD on glutamate release form cortical synaptosome. Additionally, the action of PD on inhibiting glutamate release may result from the suppression of PKC-α as well as its target SNAP-25 phosphorylation. This study provides the cellular foundations for understanding the effects of PD on glutamate and suggests a potential neuroprotective role for PD. However, the clinical relevance of our findings remains to be determined.

## 4. Materials and Methods

### 4.1. Animals and Ethics Statement

Experiments were performed after receiving permission from the Institutional Animal Care and Use Committee of the Far Eastern Memorial Hospital of Taiwan (IACUC-2022-FEMH-02). All animal procedures conformed to the Guide for the Care and Use of Laboratory Animals. Sprague–Dawley rats (male, 200–250 g; BioLASCO Co., Ltd., Taipei, Taiwan) were housed on a regular 12 light/12 dark schedule with food and water available ad libitum.

### 4.2. Chemicals, Reagents and Buffers

PD was purchased from ChemFaces (Wuhan, Hubei, China). Dantrolene, bafilomycin A1, dl-TBOA, CGP37157, PD98059, GF109203X, H89, ω-conotoxin GVIA and ω-agatoxin IVA were obtained from Tocris Cookson (Bristol, UK). 3′,3′,3′-Dipropylthiadicarbocyanine iodide (DiSC3(5)) and Fura-2-AM were purchased from Invitrogen (Carlsbad, CA, USA). EGTA). Nicotinamide adenine dinucleotide phosphate (NADP^+^), glutamate dehydrogenase (GDH), sodium dodecyl sulfate (SDS) and all other chemical reagents were supplied by Sigma-Aldrich Co. (St. Louis, MO, USA).

Sucrose buffer: 320 mM sucrose, 10 mM HEPES, pH 7.4; HEPES buffer medium (HBM): 10 mM HEPES, 140 mM NaCl, 5 mM KCl, 1 mM MgCl_2_, 5 mM NaHCO_3_, 1.2 mM NaH_2_PO_4_ and 10 mM glucose and, pH 7.4. Radioimmunoprecipitation assay buffer (RIPA): 50 mM NaCl, 1.0% IGEPAL^®^ CA-630, 0.5% sodium deoxycholate, 0.1% SDS, 50 mM Tris, pH 8.0 (Sigma-Aldrich Co., St. Louis, MO, USA). Tris-buffered saline (TBS-T): 20 mM Tris, 150 mM NaCl, 0.1% (*w*/*v*) Tween^®^ 20 detergent.

### 4.3. Synaptosomal Preparation

Synaptosomes were purified on discontinuous Percoll (Sigma-Aldrich Co., St. Louis, MO, USA) gradients, as described previously with the following modifications [25]. After the rats were decapitated, the brains were quickly removed from the skull and placed on ice. The prefrontal cortex was excised and disrupted using a Potter–Elvehjem tissue homogenizer in 10 volumes of ice-cooled sucrose buffer. The homogenate fractionation was centrifuged at 3000× *g* for 2 min at 4 °C to pellet the membrane fragments and nuclei (JA 25.5 rotor; Beckman Coulter, Inc., Brea, CA, USA). The supernatant was then transferred to a fresh tube for a second centrifugation step at 15,000× *g* for 12 min to collect the pellet containing the synaptosomes with mitochondria and microsomes. The pellet was gently resuspended in 8 mL ice-cold sucrose buffer and layered on continuous density gradients of Percoll containing 3, 10, and 23% *v*/*v* sucrose buffer. The gradients were centrifuged at 33,000× *g* for 7 min at 4 °C to collect synaptosomes located at the 10%/23% interface. To wash the samples, 30 mL of HBM was added, and the tubes were further centrifuged at 27,000× *g* for 10 min to remove Percoll. After purification, the synaptosomes were placed into HBM with 16 μM bovine serum albumin (BSA) and stored at 4 °C to remain viable for 4 h. BSA was used to remove free fatty acids released throughout treatment.

### 4.4. Glutamate Release

Glutamate release from purified prefrontal cortical synaptosomes was monitored online with an assay that employed exogenous GDH and NADP^+^ to couple the oxidative deamination of the released glutamate to the generation of NADPH detected fluorometrically [49]. A 2 mL aliquot of synaptosomes (0.5 mg of protein) was transferred to a cuvette maintained at 37 °C and stirred with a magnetic bar. Synaptosomes were preincubated with PD 10 min before depolarization. Bafilomycin A1, dl-TBOA, ω-conotoxin GVIA, ω-agatoxin IVA, dantrolene or CGP37157 were added together with PD, and protein kinase inhibitors were added 30 min before the addition of PD. After 5 min, 2 mM NADP^+^ and 50 units/mL GDH were added and allowed to stir for 3 min, and then 1 mM CaCl_2_ was added. After a further 2 min of incubation, 1.2 mM 4-AP or 15 mM KCl was added to depolarize the synaptosomes and stimulate glutamate release. The rate of increase in NADPH fluorescence at 460 nm emission (340 nm excitation) spectra was monitored over a 10 min time period using a PerkinElmer FL 6500 Fluorescence Spectrophotometer (PerkinElmer, Inc., Waltham, MA, USA). At the end of each assay, the traces were calibrated by adding 5 nmol of exogenous glutamate, and data were accumulated at 2 s intervals. The data were analyzed using GraphPad Prism 8.4.3 (GraphPad, San Diego, CA, USA), and the released glutamate was expressed as nanomoles of glutamate per milligram synaptosomal protein after 5 min of depolarization (nmol mg of protein^−1^ 5 min^−1^).

### 4.5. Measurements of Fluorometric Cytosolic Ca^2+^ ([Ca^2+^]c) and Membrane Potential

Isolated cortical synaptosomes were incubated in HBM containing CaCl_2_ (100 μM) and BSA (16 μM) with the fluorescent Ca^2+^ indicator Fura-2 AM (5 μM) and were stirred for 30 min. The synaptosome solution was then centrifuged at 5000× *g* for 5 min to remove the extracellular dye. The synaptosomes were resuspended in HBM containing 16 μM BSA and preincubated with PD (30 μM) for 10 min. In the presence of 1 mM CaCl_2_, 1.2 mM 4-AP was added to depolarize the synaptosomes. All experiments were performed at 37 °C. Relative intracellular Ca^2+^ levels were determined using a fast filter method by measuring the change in the ratio of the fluorescence excitation spectrum between 340 and 380 nm while monitoring the emission at 505 nm. Data points were collected at 2 s intervals. The [Ca^2+^]c (nM) was calculated according to the calibration procedures and equations described previously [37].

A cationic dye DiSC3(5), which can accumulate to high levels in the polarized membrane and penetrate into the lipid bilayer, was used to measure the electrical potential gradient across the membrane. The fluorescence intensity changes in DiSC3(5) were monitored at excitation and emission wavelengths of 646 and 674 nm every 2 s. Cumulative data were analyzed using GraphPad Prism and expressed in fluorescence units.

### 4.6. Western Blotting

Western blotting was performed according to the standard procedures [49]. RIPA buffer mixed with protease and phosphatase inhibitors (Thermo Fisher Scientific, Waltham, MA, USA) was used to prepare protein extracts. Protein concentrations were measured using the Pierce BCA Protein Assay Kit (Thermo Fisher Scientific, Rockford, IL, USA). Synaptosomal extracts were separated by polyacrylamide gel electrophoresis (4–12% SDS-PAGE) and subsequently transferred onto polyvinylidene difluoride (PVDF) membranes (Bio-Rad Laboratories, Inc., Hercules, CA, USA). The membranes were blocked with TBS-T containing 5% BSA. The following primary antibodies were used: rabbit polyclonal antibodies directed against phospho-PKC-α, phospho-SNAP-25 and β-actin (Cell Signaling Technology, Beverly, MA, USA). Horseradish peroxidase-conjugated donkey anti-rabbit secondary antibodies (BioLegend, Inc., San Diego, CA, USA) were used to detect the respective primary antibodies. Immunoreactive proteins were developed with Amersham ECL prime Western blotting detection reagent (Cytiva, Marlborough, MA, USA). Chemiluminescent signals were detected with an ImageQuant™ LAS-4000 imaging system (Fujifilm, Tokyo, Japan). The level of expression or phosphorylation was quantified using Multi Gauge V3.0 (Fuji Photo Film. Co. Ltd., Mini-ku, Tokyo, Japan).

### 4.7. Statistical Analysis

All graphs, calculations and statistical analyses were performed using Prism software (version 8.4.3, GraphPad, San Diego, CA, USA). Data are expressed as the means with standard error of the means (SEMs) for 3–5 independent experiments. A two-tailed Student’s t test was used to compare the significance of the effect of a drug versus the control, whereas a one-way repeated-measures analysis of variance (ANOVA) was used for multiple comparisons among three or more different groups. Tukey’s honest significant difference (HSD) was used as a post hoc test to determine significant differences between groups. The differences were considered significant when *p* < 0.05.

## Figures and Tables

**Figure 1 molecules-28-01313-f001:**
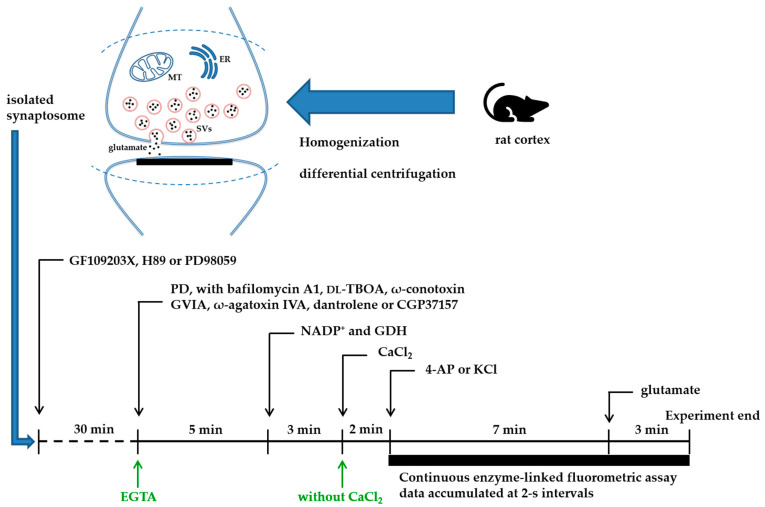
Schematic diagram of the experimental design and treatment schedule. EGTA treatment is represented in green. Abbreviations: MT, mitochondria; ER, endoplasmic reticulum; SVs, synaptic vesicles.

**Figure 2 molecules-28-01313-f002:**
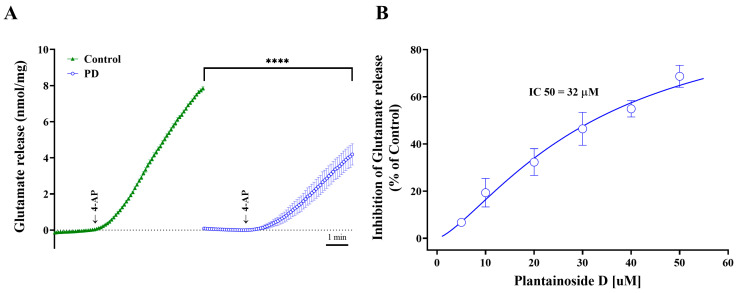
PD inhibits 4-AP-induced glutamate release from rat cerebrocortical nerve terminals. (**A**) Representative online measurement of glutamate release from synaptosomes evoked by the potassium channel blocker 4-AP in the presence of 30 μM PD. As described in the Materials and Methods Section, synaptosomes (0.5 mg of protein) were resuspended in HBM, transferred to stirred cuvettes and incubated for 10 min before 1.2 mM 4-AP was added. Using 50 units/mL glutamate dehydrogenase (GDH) in the presence of 2 mM NADP^+^, glutamate release was measured in the absence (control conditions) or presence of 30 μM PD, which was added 10 min before 4-AP was added. The experiments were performed in the presence of 1 mM CaCl_2_. (**B**) PD inhibits 4-AP-evoked glutamate release in a dose-dependent manner, with an IC50 value of 32 μM. Each dot represents five independent experiments of online glutamate release measurement and are calculated as the means ± standard errors of the means (SEMs) (*n* = 5 per group). **** *p <* 0.0001.

**Figure 3 molecules-28-01313-f003:**
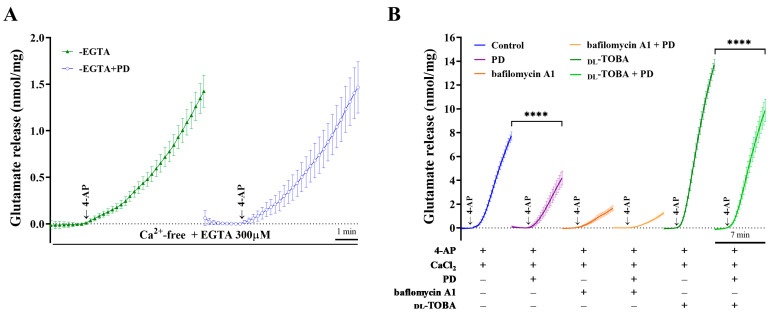
Determining the effects of the Ca^2+^ chelator EGTA, the nonselective glutamate transporter inhibitor dl-TBOA, and the cellular autophagy inhibitor bafilomycin A1 on the inhibitory effect of PD. (**A**) Ca^2+^ deprivation induced by excluding Ca^2+^ and the addition of EGTA, a Ca^2+^ chelator, showed no effect on PD-mediated inhibition of glutamate release. Each point represents data collected at 2 s intervals. For clarity, only the time point and error bars every 10 s are shown. (**B**) Isolated synaptosome were divided into 6 experimental groups as follows: (i) control, (ii) PD, (iii) bafilomycin A1, (iv) bafilomycin A1 + PD, (v) dl-TBOA and (vi) dl-TBOA + PD. It shows the inhibitory effect of 30 μM PD, added 10 min before depolarization, on 4AP-evoked glutamate release in the presence or in the absence of 10 μM dl-TBOA or 0.1 μM bafilomycin A1. The results shown are averages of five independent experiments from isolated synaptosomes and represent the means ± SEMs (*n* = 5 per group). **** *p <* 0.0001 versus controls or dl-TBOA treatment.

**Figure 4 molecules-28-01313-f004:**
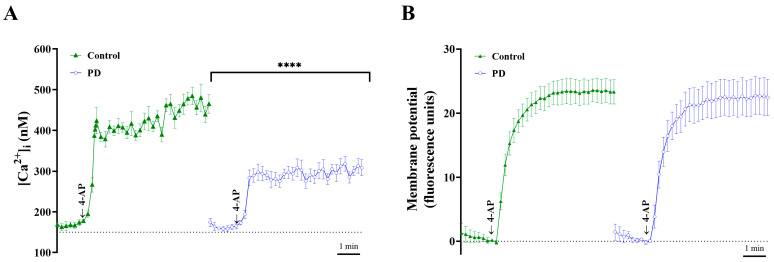
PD attenuates 4-AP-evoked voltage-dependent Ca^2+^ influx but shows no effect on the synaptosomal membrane potential. (**A**) lntrasynaptosomal levels of Ca^2+^ were measured using the fluorescent indicator Fura-2 (5 μM). PD (30 μM) was added 10 min before depolarization with 1 mM 4-AP, and [Ca^2+^]c was analyzed by spectrometry. (**B**) Depolarization of synaptosomes with 4-AP. The membrane potential was monitored with 5 μM DiSC3(5) in the absence (control) or presence of 30 μM PD. The results are shown as the averages of five independent experiments, and the data represent the means ± SEMs (*n* = 5 per group). **** *p <* 0.0001.

**Figure 5 molecules-28-01313-f005:**
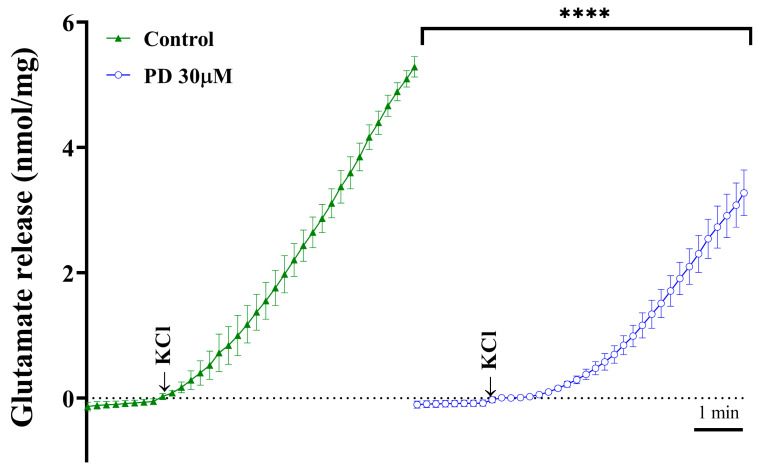
Effect of PD on 15 mM KCl-evoked glutamate release. The results are shown as the averages of five independent experiments, and the data represent the means ± SEMs (*n* = 5 per group). **** *p <* 0.0001 versus the control.

**Figure 6 molecules-28-01313-f006:**
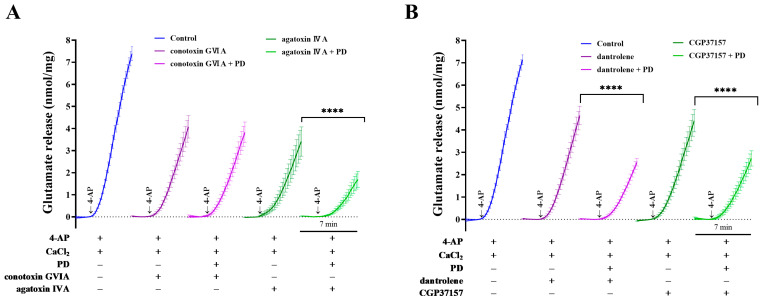
Sensitivity of the PD-mediated inhibition of 4-AP-evoked glutamate release to the N-type VDCC blocker ω-conotoxin GVIA (0.5 μM), P/Q-type VDCC blocker ω-agatoxin IVA (0.1 μM), and inhibitors of the mitochondrial Na^+^/Ca^2+^ exchanger and ryanodine receptor (CGP37157 and dantrolene). (**A**) Isolated synaptosome were divided into 5 experimental groups as follows: (i) control, (ii) ω-conotoxin GVIA, (iii) ω-conotoxin GVIA + PD, (iv) ω-agatoxin IVA and (v) ω-agatoxin IVA + PD. PD inhibits 4AP-induced Ca^2+^ entry into nerve terminals through N-type voltage-dependent VDCCs. Each point collected at 2 s intervals represents the data of glutamate release induced by 4-AP in the absence (control) or presence of 30 μM PD. The time points and error bars are shown once for every 10 s for clarity. (**B**) Isolated synaptosome were divided into 5 experimental groups as follows: (i) control, (ii) dantrolene, (iii) dantrolene + PD, (iv) CGP37157 and (v) CGP37157 + PD. PD application maintained the inhibitory effect on glutamate release in the presence of dantrolene (10 μM) or CGP37157 (10 μM). Data are presented as the means ± SEMs of 5 independent experiments from isolated synaptosomes (*n* = 5 per group). **** *p <* 0.0001 compared with ω-agatoxin IVA, dantrolene or CGP37157 treatment.

**Figure 7 molecules-28-01313-f007:**
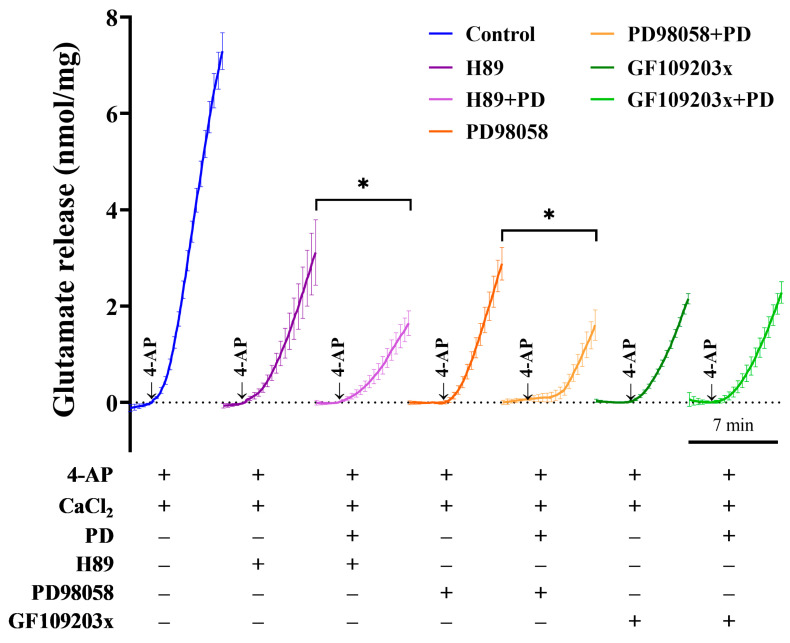
The PKC pathway is responsible for the PD-mediated inhibition of glutamate release. Isolated synaptosome were divided into 7 experimental groups as follows: (i) control, (ii) H89, (iii) H89 + PD, (iv) PD98059, (v) PD98059 + PD, (vi) GF109203X and (vii) GF109203X + PD. The PD-mediated inhibition of 4AP-evoked glutamate release is abolished by the PKC inhibitor GF109203X (5 μM) but not the PKA inhibitor H89 (10 μM) or MAPK/ERK inhibitor PD98059 (20 μM). Data are presented as the means ± SEMs of 5 independent experiments from isolated synaptosomes (*n* = 5 per group). * *p <* 0.01 versus H89 or PD98059 treatment.

**Figure 8 molecules-28-01313-f008:**
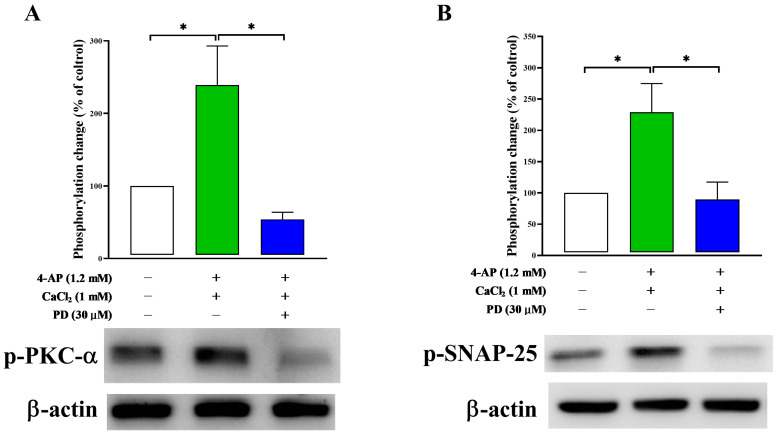
Effect of PD on 4-AP-evoked phosphorylation of PKC-α (**A**) and SNAP-25 (**B**), which is a target of PKC. PD was added 10 min before depolarization with 4-AP. The phosphorylation level was measured at 10 min after 4-AP depolarization and expressed as a percentage of that measured in the control group without 4-AP. Each bar represents the means ± SEMs of the results obtained in 3 experiments (*n* = 3 per group). * *p <* 0.05 versus the control or 4-AP-treated group.

## Data Availability

The data presented in this study are available upon request from the corresponding author.

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
