# Peer review of "Plantainoside D Reduces Depolarization-Evoked Glutamate Release from Rat Cerebral Cortical Synaptosomes"

_molecules, 2023, doi:10.3390/molecules28031313_

Round 1

Reviewer 1 Report

The authors presented a highly valuable research concerning the evaluation of the effectiveness of Plantainoside D; a natural product, for decreasing the excessive release of the deletrious excitatory neurotransmitters; glutamate in the brain cortical area in a rate model. It is a very important and delicate research. In my opinion, it may gain a great attention to the audience after publication.  

Meanwhile, I have very important points that may improve the marketing of the manuscript:

1- Improve the Materials and Methods section and be more specific for the grouping and animal treatments

A. The method of treatment of groups of rats and the number of rats in each group must be mentioned.

B. The Dose of the EGTA has been mentioned to be 300 micrMol. It has to be verified in the dose response curve. It has no reference for the reason of using such dose (reference number 24 is not include EGTA)

C. I need a verification of the sequence and the time relapse between treatment using EGTA and Plantainoside D. There may be a complexation between them or interference for their effect on calcium during treatment. Kindly explain this in the discussion section.

D. Kindly add a recommendation for the usage of Plantainoside D for treatment or the usage with a recommended dosage and recommended neurodegenerative diseases.

E. I prefer to add a figure showing the treatment and the duration of each treatment in order to follow the treatment in each group

2. In the result section

A. If you have histopathological sections, it will dramatically improve the output of using such important active constituents.

It will show the damage effect of treatments and the protective effects on the brain tissues very clearly

2- Update the references section

A. Some references such as 20, 21, 22, 23, 24, 24, 26, 27, 28, 36, 38, 42, 43, 46 and 48 are very old. Kindly update these references not before 2017. 

Author Response

molecules-2152632R1

We thank the reviewers for the critical comments and constructive suggestions.

Reviewer 1

The authors presented a highly valuable research concerning the evaluation of the effectiveness of Plantainoside D; a natural product, for decreasing the excessive release of the deletrious excitatory neurotransmitters; glutamate in the brain cortical area in a rate model. It is a very important and delicate research. In my opinion, it may gain a great attention to the audience after publication. 

Meanwhile, I have very important points that may improve the marketing of the manuscript:

1- Improve the Materials and Methods section and be more specific for the grouping and animal treatments

  1. The method of treatment of groups of rats and the number of rats in each group must be mentioned.

As suggestion by the reviewer, the method of treatment of groups and the number of rats in each group are added (page 4, line 121; page 5, line 152-154, 157-158; page 6, line 189; page 7, line 220; page 8, line 227-229, 232-233, 236; page 9, line 257-258, 260; page 10, line 281; page 13, line 430-432).

  1. The Dose of the EGTA has been mentioned to be 300 micrMol. It has to be verified in the dose response curve. It has no reference for the reason of using such dose (reference number 24 is not include EGTA)

The dose of the EGTA has been shown in Figure 2A. The reference for the reason of using this dose has been added (reference 25, page 4, line 128).

  1. I need a verification of the sequence and the time relapse between treatment using EGTA and Plantainoside D. There may be a complexation between them or interference for their effect on calcium during treatment. Kindly explain this in the discussion section.

As suggestion by the reviewer, the sequence and the time relapse between treatment of EGTA and PD is shown in figure 1. In order to make the statement of the sentence clearer, several sentences are added in the discussion section (page 11, lines 306-309). “In Ca2+-free medium, EGTA was added to each preparation together with PD, followed after 10 min by the addition of 4-AP. PD did not result in a significant decrease in Ca2+-independent glutamate release, which depends solely on the membrane potential.”

  1. Kindly add a recommendation for the usage of Plantainoside D for treatment or the usage with a recommended dosage and recommended neurodegenerative diseases.

In order to make the statement of the sentence clearer, several sentences are added in the discussion section (page 12, line 369-375). “In this study, the ability of PD to decrease glutamate release is dose dependent with IC50 value of 32 μM. Consistent with our study, 20 μg/ml PD inhibits cardiac muscle cell apoptosis. In addition, in an in vivo animal study, PD was administered intravenously (2.0 mg/kg) and orally (10 mg/kg) to rats for pharmacokinetic analysis. They found that single dose oral administration of PD results in a low bioavailability in vivo. Since there are very few in vivo studies of PD in literature, further studies are needed to understand the dosage for the treatment of neurodegenerative diseases.”

  1. I prefer to add a figure showing the treatment and the duration of each treatment in order to follow the treatment in each group

 As suggestion by the reviewer, figure 1 is added (Page3, line 94-95, 104-107). Several sentences “Bafilomycin A1, dl-TBOA, ω-conotoxin GVIA, ω-agatoxin IVA, dantrolene or CGP37157 were added together with PD, and protein kinase inhibitors were added 30 min before the addition of PD.” are added in the materials and methods section (Page 13, line 430-432).

  1. In the result section
  2. If you have histopathological sections, it will dramatically improve the output of using such important active constituents.

It will show the damage effect of treatments and the protective effects on the brain tissues very clearly

We agree this point mentioned by the reviewer. However, this part of the experiment cannot be performed in the present study due to the limited response time. Thank you very much for the suggestion of reviewer. We will pay attention to this point in the future work.

2- Update the references section

  1. Some references such as 20, 21, 22, 23, 24, 25, 26, 27, 28, 36, 38, 42, 43, 46 and 48 are very old. Kindly update these references not before 2017.

References 22, 23, 24, 25, 26, 27, 28, 36, 38, 42, 43, 46 and 48 were updated.

The references 20, 21 were not updated since these were no similar references in literature.

Reviewer 2 Report

The authors tried to investigate the effect and mechanism of plantainoside D (PD) on glutamate release in rat synaptosomes. Their results suggest that PD mediates the inhibition of evoked glutamate release from synaptosomes primarily by reducing the influx of Ca2+ through N-type Ca2+ channels, subsequently reducing the protein kinase C cascade. It is interesting, and some comments are as follows:

1.    In this manuscript, the authors suggest that the PKC pathway is responsible for the PD-mediated inhibition of glutamate release. In Figure 7, they evaluated the phosphorylation of PKC-a,and SNAP25,how about the other molecules, such as MAPK/ERK, and PKA? And what is the relationship between p-SNAP25 and PKC? In figure 7, it was suggested to demonstrate the total proteins of PKC-a and SNAP25 and their relationship.

 2. There are several subtypes of PKC, and which subtype is primary to regulate the release of glutamate? Could PKC-a represent the function of PKC family? 

Author Response

molecules-2152632R1

We thank the reviewers for the critical comments and constructive suggestions.

Reviewer 1

Comments and Suggestions for Authors

The authors tried to investigate the effect and mechanism of plantainoside D (PD) on glutamate release in rat synaptosomes. Their results suggest that PD mediates the inhibition of evoked glutamate release from synaptosomes primarily by reducing the influx of Ca2+ through N-type Ca2+ channels, subsequently reducing the protein kinase C cascade. It is interesting, and some comments are as follows:

  1. In this manuscript, the authors suggest that the PKC pathway is responsible for the PD-mediated inhibition of glutamate release. In Figure 7, they evaluated the phosphorylation of PKC-a,and SNAP25,how about the other molecules, such as MAPK/ERK, and PKA? And what is the relationship between p-SNAP25 and PKC? In figure 7, it was suggested to demonstrate the total proteins of PKC-a and SNAP25 and their relationship.

Regarding to this point, we examined the effect of the PKA inhibitor H89 and MAPK/ERK inhibitor PD98059 on PD-mediated inhibition of glutamate release. The inhibitory effect of PD on the 4- AP-evoked glutamate release was not affected by H89 or PD98059. These results suggest that the PKA or MAPK/ERK pathway is not responsible for the PD-mediated inhibition of glutamate release (Figure 7).

In order to make the statement of the sentence clearer, the sentences in the discussion section are modified (page 12, line 359-364; page12, line 366-367). “The synaptic protein SNAP-25 is a target of PKC and a member of the SNARE family of vesicular fusion proteins. PKC phosphorylation of residue serine 187 (Ser 187) in SNAP-25 is important for assembly of the SNARE complex and Ca2+-triggered synaptic vesicle exocytosis.” “In our previous study, no significant difference was observed in total proteins of PKC-α and SNAP25 between the groups.”

  1. There are several subtypes of PKC, and which subtype is primary to regulate the release of glutamate? Could PKC-a represent the function of PKC family?

In order to make the statement of the sentence clearer, several sentences are added in the discussion section (page 11, line 353-357; page 12, line 358-359). “However, in addition to PKC-α the possible involvement of other PKC isoforms could not be ruled out. For instance, PKC-β (β1 and β2) has been reported to be involved in many different cellular functions with a Ca2+-dependent manner.”